# Machine Learning Consensus Clustering Approach for Hospitalized Patients with Phosphate Derangements

**DOI:** 10.3390/jcm10194441

**Published:** 2021-09-27

**Authors:** Charat Thongprayoon, Carissa Y. Dumancas, Voravech Nissaisorakarn, Mira T. Keddis, Andrea G. Kattah, Pattharawin Pattharanitima, Tananchai Petnak, Saraschandra Vallabhajosyula, Vesna D. Garovic, Michael A. Mao, John J. Dillon, Stephen B. Erickson, Wisit Cheungpasitporn

**Affiliations:** 1Division of Nephrology and Hypertension, Department of Medicine, Mayo Clinic, Rochester, MN 59005, USA; dumancas.carissa@mayo.edu (C.Y.D.); kattah.andrea@mayo.edu (A.G.K.); garovic.Vesna@mayo.edu (V.D.G.); dillon.John@mayo.edu (J.J.D.); erickson.stephen@mayo.edu (S.B.E.); 2Division of Nephrology, Department of Medicine, Beth Israel Deaconess Medical Center, Harvard Medical School, Boston, MA 02215, USA; voravech.niss@gmail.com; 3Division of Nephrology and Hypertension, Department of Medicine, Mayo Clinic, Phoenix, AZ 85054, USA; keddis.Mira@mayo.edu; 4Department of Internal Medicine, Faculty of Medicine, Thammasat University, Pathum Thani 12120, Thailand; 5Division of Pulmonary and Pulmonary Critical Care Medicine, Faculty of Medicine, Ramathibodi Hospital, Mahidol University, Bangkok 10400, Thailand; petnak@yahoo.com; 6Section of Cardiovascular Medicine, Department of Medicine, Wake Forest University School of Medicine, Winston-Salem, NC 27101, USA; svallabh@wakehealth.edu; 7Division of Nephrology and Hypertension, Mayo Clinic, Jacksonville, FL 32224, USA; mao.michael@mayo.edu

**Keywords:** phosphate, hyperphosphatemia, hypophosphatemia, machine learning, artificial intelligence, clustering, electrolytes, nephrology, precision medicine, personalized medicine, individualized medicine

## Abstract

Background: The goal of this study was to categorize patients with abnormal serum phosphate upon hospital admission into distinct clusters utilizing an unsupervised machine learning approach, and to assess the mortality risk associated with these clusters. Methods: We utilized the consensus clustering approach on demographic information, comorbidities, principal diagnoses, and laboratory data of hypophosphatemia (serum phosphate ≤ 2.4 mg/dL) and hyperphosphatemia cohorts (serum phosphate ≥ 4.6 mg/dL). The standardized mean difference was applied to determine each cluster’s key features. We assessed the association of the clusters with mortality. Results: In the hypophosphatemia cohort (*n* = 3113), the consensus cluster analysis identified two clusters. The key features of patients in Cluster 2, compared with Cluster 1, included: older age; a higher comorbidity burden, particularly hypertension; diabetes mellitus; coronary artery disease; lower eGFR; and more acute kidney injury (AKI) at admission. Cluster 2 had a comparable hospital mortality (3.7% vs. 2.9%; *p* = 0.17), but a higher one-year mortality (26.8% vs. 14.0%; *p* < 0.001), and five-year mortality (20.2% vs. 44.3%; *p* < 0.001), compared to Cluster 1. In the hyperphosphatemia cohort (*n* = 7252), the analysis identified two clusters. The key features of patients in Cluster 2, compared with Cluster 1, included: older age; more primary admission for kidney disease; more history of hypertension; more end-stage kidney disease; more AKI at admission; and higher admission potassium, magnesium, and phosphate. Cluster 2 had a higher hospital (8.9% vs. 2.4%; *p* < 0.001) one-year mortality (32.9% vs. 14.8%; *p* < 0.001), and five-year mortality (24.5% vs. 51.1%; *p* < 0.001), compared with Cluster 1. Conclusion: Our cluster analysis classified clinically distinct phenotypes with different mortality risks among hospitalized patients with serum phosphate derangements. Age, comorbidities, and kidney function were the key features that differentiated the phenotypes.

## 1. Introduction

Phosphate is an essential element in the body and the most abundant intracellular anion, with the majority of it stored in bones, and < 1% found in the serum [1,2,3,4,5]. Normal serum phosphate levels are maintained within a relatively narrow range (2.5–4.5 mg/dL) because they are vital in cellular metabolism, signal transduction, and bone homeostasis [6,7,8,9,10]. Abnormal serum phosphate levels are common, with a prevalence up to 45% in hospitalized patients [11,12], and are associated with poor clinical outcomes, including cardiovascular disease [2,3], respiratory failure [13], and increased mortality [7,11,14,15,16,17].

The application of machine learning (ML) to medicine recently became more popular for individualized medicine [18]. The concept of ML is the ability of computers to distinguish and analyze trends or patterns in data to make predictions without explicitly being programmed [19]. The use of ML to process large and complex data from electronic health records (EHRs) has led to advances in precision medicine [20]. Unsupervised ML techniques have identified novel data patterns and distinct subtypes in different diseases [21,22,23,24,25]. It can identify similarities and differences among multiple data variables and divide them into meaningful clusters [21,22]. Furthermore, previous investigations have indicated that distinct subtypes identified by ML consensus clustering algorithms are associated with different clinical outcomes [26,27]. Given that hospitalized patients with abnormal serum phosphate are heterogeneous, ML can find distinct clusters with different clinical outcomes. Identifying these distinct clusters may be beneficial if it can change the approach to understanding the characteristics of patients with phosphate disorders upon hospital admission and their associated mortality risks.

This study aimed to use an unsupervised ML consensus clustering algorithm to classify distinct clusters of hospitalized patients with abnormal serum phosphate upon admission and assess the associated mortality risk in the identified phenotypes.

## 2. Methods

### 2.1. Patient Population

The Mayo Clinic Institutional Review Board approved this study (IRB number 21-003088 and date of approval; 30 March 2021), and all included patients provided research authorization. We screened adult patients (age ≥ 18 years) admitted to the Mayo Clinic Hospital, Rochester, Minnesota, USA, from January 2009 to December 31, 2013. We included patients who presented with abnormal admission serum phosphate outside the normal reference range (2.5–4.6 mg/dL). We excluded patients who did not have a serum phosphate measurement within 24 h of hospital admission, or that had normal admission serum phosphate. We separated patients into two cohorts: (1) the hypophosphatemia cohort (serum phosphate ≤ 2.4 mg/dL), and (2) the hyperphosphatemia cohort (serum phosphate ≥ 4.6 mg/dL).

### 2.2. Data Collection

We used our hospital’s electronic database to abstract pertinent demographic information, comorbidities, principal diagnoses, and laboratory data, as previously described. The cluster analysis only utilized data available within 24 h of hospital admission. The initial laboratory value was selected for use if multiple values were available within the 24-h time frame. We excluded variables with over 10% missing data. If a variable had absent data of less than 10%, we imputed the missing data using the random forest multiple imputation technique before inputting the data into the cluster analysis. We used the missForest package for missing data imputation.

The outcomes were hospital, one-year, and five-year mortality. Patient death was obtained from our hospital’s registry and the Social Security Death Index. The last follow-up date was 31 December 2018. The median follow-up date was 6.1 (IQR 1.8–8.0) years.

### 2.3. Cluster Analysis

Unsupervised ML consensus clustering analysis was applied to identify clinical clusters of hypophosphatemia and hyperphosphatemia cohorts [28]. We utilized a prespecified subsampling parameter of 80% with 100 iterations. The number of possible clusters (k) was selected to be between 2 and 10 to avoid disproportionate numbers of clusters that are not clinically meaningful. The ideal number of clusters was ascertained by evaluating the cumulative distribution function (CDF), the consensus matrix (CM) heat map, cluster-consensus plots, and the proportion of ambiguously clustered pairs (PAC) [29,30]. The within-cluster consensus score is defined as the mean consensus value for all pairs of individuals belonging to the same cluster (range 0–1) [30]. A value that is closer to 1 represents higher cluster stability [30]. The PAC (range 0–1) is calculated as the proportion of all sample pairs with consensus values that fall within the predetermined boundaries [29]. A value that is closer to 0 signifies higher cluster stability [29]. The details regarding the consensus cluster algorithms can be found in the Appendix A.

### 2.4. Statistical Analysis

After cluster identification, we performed analyses to characterize differences among the clusters. First, we compared the baseline characteristics between the clusters using the Student’s t-test for continuous variables, and the Chi-squared test for categorical variables. We used the standardized mean difference of the clinical characteristics between each cluster and the overall cohort to determine the key features of each cluster. A clinical characteristic with an absolute standardized mean difference of > 0.3 represented a key feature for each cluster. Then, we compared hospital mortality and one-year mortality between the clusters. We evaluated the association of the cluster with hospital mortality using logistic regression and the reported odds ratio (OR) with a 95% confidence interval (95% CI). We evaluated the association of the cluster with one-year mortality using Cox proportional hazard regression and the reported hazard ratio (HR) with a 95% CI. We did not adjust for differences in the clinical variables between the groups because these variables were utilized through unsupervised machine learning to identify the clusters. We used the ConsensusClusterPlus package (version 1.46.0) (https://bioconductor.org/packages/release/bioc/html/ConsensusClusterPlus.html; accessed on 15 July 2021) for the consensus clustering analysis. We used R, version 4.0.3 (RStudio, Inc., Boston, MA, USA), for all analyses.

## 3. Results

### 3.1. Hypophosphatemia Cohort

There were 41,273 hospitalized patients with available admission serum phosphate measurements. Of these, 3113 (8%) patients presented with hypophosphatemia upon hospital admission. The mean age was 61 ± 17 years. 53% were male. The mean estimated glomerular filtration rate (eGFR) was 82 ± 29. The mean admission serum phosphate was 2.0 ± 0.4 mg/dL (Table 1).

The CDF plot demonstrates the consensus distributions for each hypophosphatemia cluster (Appendix A). The delta area plot, in turn, demonstrates the relative change in area under the CDF curve (Appendix A). The most significant changes in area occurred between k = 2 and k = 4. Beyond this range, the relative increment in area was significantly smaller. The CM heatmap (Figure 1A, Appendix A) reveals that the clustering algorithm identified two clusters with sharp boundaries (Figure 1A), representing excellent cluster stability over repeated iterations. Cluster 2 also had the highest mean cluster consensus score, representing high stability (Figure 2A). Favorable low PACs were demonstrated for two clusters (Appendix A). Thus, the consensus clustering analysis from available hospital admission baseline characteristics identified two clusters that best represented the data pattern of our patients admitted with hypophosphatemia.

Cluster 1 had 1505 (48%) patients, while Cluster 2 had 1608 (52%) patients. As shown in Table 1, the clinical characteristics between the two identified clusters in the hypophosphatemia cohort were significantly different. On the basis of the standardized mean difference shown in Figure 3, the key features of patients in Cluster 2, compared with Cluster 1, included: older age; a higher comorbidity burden, in particular hypertension; diabetes mellitus; coronary artery disease; lower eGFR; more acute kidney injury (AKI) at admission; more use of angiotensin converting enzyme inhibitors (ACEI)/angiotensin receptor blockers (ARB); and diuretics before admission.

Cluster 1 had a hospital mortality of 2.9%, whereas Cluster 2 had a hospital mortality of 3.7% (*p* = 0.17) (Figure 4A). There was no difference in hospital mortality between Cluster 1 and Cluster 2. In contrast, Cluster 1 had a one-year mortality of 14.0% and a five-year mortality of 20.2%, whereas Cluster 2 had a one-year mortality of 26.8% and a five-year mortality of 44.3% (*p* < 0.001) (Figure 4B).

Cluster 2 had a higher one-year mortality and five-year mortality, when compared to Cluster 1, with HRs of 2.10 (95% CI 1.75–2.52), and 2.56 (95% CI 2.24–2.93), respectively (Table 2a).

### 3.2. Hyperphosphatemia Cohort

A total of 7252 patients presented with hyperphosphatemia upon hospital admission. The mean age was 60 ± 18 years. 55% were male. The mean eGFR was 55 ± 39. The mean admission serum phosphate was 5.5 ± 1.2 mg/dL (Table 1).

The CDF plot demonstrates the consensus distributions for each hyperphosphatemia cluster (Appendix A). The delta area plot, in turn, demonstrates the relative change in area under the CDF curve (Appendix A). The most significant changes in area occurred between k = 2 and k = 4. Beyond this range, the relative increment in area was significantly smaller. The CM heatmap (Figure 1B, Appendix A) reveals that the clustering algorithm identified Cluster 2 with sharp boundaries (Figure 1B), representing excellent cluster stability over repeated iterations. Cluster 2 also had the highest mean cluster consensus score, representing high stability (Figure 2B). Favorable low PACs were demonstrated for two clusters (Appendix A). Thus, the consensus clustering analysis from the available hospital admission baseline characteristics identified two clusters that optimally represented the data pattern of our patients admitted with hyperphosphatemia.

Cluster 1 had 3662 (51%) patients, while Cluster 2 had 3590 (49%) patients. As shown in Table 1, the clinical characteristics between the two identified clusters in the hyperphosphatemia cohort were significantly different. Based on the standardized mean difference shown in Figure 3, the key features of patients in Cluster 2, when compared with Cluster 1, included: older age; more primary admission for kidney disease; more history of hypertension, and end-stage kidney disease; more AKI at admission; more use of diuretics; and higher admission potassium, magnesium, and phosphate.

Cluster 1 had a hospital mortality of 2.4%, whereas Cluster 2 had a hospital mortality of 8.9% (*p* < 0.001) (Figure 4C). Cluster 2 had a higher hospital mortality when compared with Cluster 1, with an OR of 4.06 (95% CI 3.18–5.17). Similarly, Cluster 1 had a one-year mortality of 14.8%, and a five-year mortality of 24.5%, whereas Cluster 2 had a one-year mortality of 32.9%, and a five-year mortality of 51.1% (*p* < 0.001) (Figure 4D). Cluster 2 had a higher one-year and five-year mortality when compared with Cluster 1, with HRs of 2.63 (95% CI 2.36–2.93), and 2.58 (95% CI 2.38–2.79), respectively (Table 2b).

## 4. Discussion

The unsupervised ML consensus clustering approach offers the ability to more efficiently analyze, identify, and classify groups of patients based on phenotypic features in large volumes of data. [21,22,23,24] In this study, the unsupervised ML consensus clustering algorithm was applied to classify patients with phosphate disorders into unique clusters. Age, comorbidities, and kidney function were the important features used to differentiate the phenotypes of phosphate disorders upon hospital admission, both hypophosphatemia and hyperphosphatemia. These produced clusters of phosphate disorders had high cluster stability, with different patients’ characteristics. In addition, these distinct clusters were also associated with different hospital and one-year mortality risks.

The kidney is an important regulator of phosphorus homeostasis, and hypophosphatemia can be caused by poor phosphorus intake/intestinal absorption, redistribution from extracellular to intracellular compartments (refeeding syndrome and respiratory alkalosis), and/or excessive urinary phosphate excretion [7,16,31]. Upon hospital admission, the findings of our ML consensus clustering suggest that kidney functions (surrogates included baseline eGFR and AKI on admission) played an important role in differentiating the phenotypes of patients with hypophosphatemia.

Patients in Cluster 1 of hypophosphatemia had a higher baseline eGFR and a lower incidence of AKI. Compared with Cluster 2, patients in Cluster 1 were younger and had fewer medical comorbidities. These patients had a higher history of alcohol use and had more principal diagnosis of injury. Underlying mechanisms of hypophosphatemia in alcoholic patients include inappropriate renal phosphate excretion, enhanced cellular uptake of phosphorus, and decreased gastrointestinal phosphate absorption [31,32]. Alcoholic patients commonly have reduced serum calcium, phosphate, and potassium levels [32], as was also demonstrated in our patients in Cluster 1. Furthermore, the majority of these patients also had hypoalbuminemia. Thus, it is possible that alcoholism played an important role in the development of hypophosphatemia in this patient population.

Conversely, Cluster 2 of hypophosphatemia, compared with Cluster 1, included: older age; lower eGFR; more AKI at admission; a higher comorbidity burden, particularly hypertension; diabetes mellitus; and coronary artery disease. While patients with reduced kidney functions commonly have hyperphosphatemia [9], hypophosphatemia can also be found in patients with CKD or AKI upon hospital admission, especially among those with malnutrition [16]. Compared with Cluster 1, Cluster 2 had more principal diagnoses of cardiovascular, genitourinary, and hematoma oncology. Hypophosphatemia in Cluster 2 could be a surrogate marker of illness, or related to comorbidities and their treatment [7,33]. For instance, diabetes mellitus being treated with insulin therapy is another common comorbidity that is associated with hypophosphatemia. In addition, among patients with cardiovascular diseases, hypophosphatemia may be the result of coexisting alkalosis, pharmacological treatments (such as diuretics), a reduced intestinal absorption of phosphate, or secondary to sympathetic nervous system activation [34].

Despite the conflicting data on the impacts of hypophosphatemia on patient survival [12,35,36,37,38,39,40,41,42,43], when compared with patients with normal phosphate levels, studies have demonstrated that hypophosphatemia is also associated with increased hospital mortality [12,16,35,36,37,38,39,40,41,42,43]. Furthermore, severe hypophosphatemia has been reported to cause rhabdomyolysis, respiratory failure, and metabolic encephalopathy [4,31]. Our current study additionally assessed the mortality risks among these two clusters of hypophosphatemia with different phenotypes. While we found a comparable in-hospital mortality risk among these two clusters, the patients in Cluster 2 carried a higher one-year mortality compared to Cluster 1, despite having less severity of hypophosphatemia. This is likely due to the effects of old age and comorbidities.

Causes of hyperphosphatemia include: decreased GFR; exogenous phosphate sources (phosphate supplement, phosphate enemas, high phosphate diet): an endogenous load of phosphate (tumor lysis syndrome and rhabdomyolysis); and increased tubular phosphate reabsorption [9,16]. As demonstrated in the ML consensus clustering analysis of hypophosphatemia, the phenotypes of patients with hyperphosphatemia upon admission were also mainly influenced by baseline eGFR and AKI upon admission, dividing hyperphosphatemic patients into two clusters. Given that urinary phosphate excretion is the key mechanism in maintaining serum phosphate levels [11], decreased GFR in patients with AKI or CKD can result in hyperphosphatemia [44,45]. Conversely, acute hyperphosphatemia itself can result in AKI from acute phosphate nephropathy [46,47,48,49]. The key features of the patients in Cluster 2 included: older age; more history of hypertension; reduced eGFR; more AKI at admission; and higher admission potassium, magnesium, and phosphate levels, when compared with Cluster 1. Primary admission for genitourinary, mainly kidney failure, is an important feature of Cluster 2, while patients in Cluster 1 had higher primary admission for hematoma/oncology. Cluster 2 had both a higher hospital and one-year mortality compared with Cluster 1. While both the short- and long-term mortality among patients in Cluster 2 could have been the result of the effects of older age, comorbidities, and AKI [50], these patients also had a higher degree of hyperphosphatemia, which was associated with worse clinical outcomes, including cardiovascular events and mortality [51,52,53,54].

There were several limitations to our current study. First, the data from our study are from a single center that may be unique to our patient population (the predominant population in our study was Caucasian). Second, the ML clustering approach was conducted at the time of hospital admission to allow application of this study to clinical practice and future studies. Future studies should include an evaluation as to whether the early recognition of mortality risk in hospitalized patients with distinct phenotypes of phosphate disorders would permit earlier intervention and the mitigation of mortality. Third, some laboratory investigations that may have affected the phosphate levels or helped to determine the causes of phosphate disorders, were not commonly performed upon admission (urine phosphate excretion, parathyroid hormone, 25-hydroxyvitamin D level, arterial blood gases, fibroblast growth factor 23 (FGF23)) and, thus, were not included in our ML clustering algorithm. Lastly, data on medications that can alter phosphate levels, such as insulin, phosphate supplement, phosphate binders, were limited in our database. Thus, future studies are needed to assess whether these variables could have improved the discriminatory ability of the clusters we identified. Nevertheless, our ML clustering approach successfully identified clusters with distinct phenotypes among hospitalized patients with phosphate disorders that indicated different mortality risks.

## 5. Conclusions

ML consensus clustering analysis identified distinct clusters of hospitalized patients with admission phosphate disorders. Age, comorbidities, and kidney function were the key features used to differentiate the phenotypes of phosphate disorders upon hospital admission. Furthermore, the distinct phenotypes of phosphate disorders have differing in-hospital and one-year mortality risks. Future studies on interventional targets to improve the outcomes of phosphate disorders may be potentially beneficial for focusing on patients with phenotypes of high mortality risks.

## Figures and Tables

**Figure 1 jcm-10-04441-f001:**
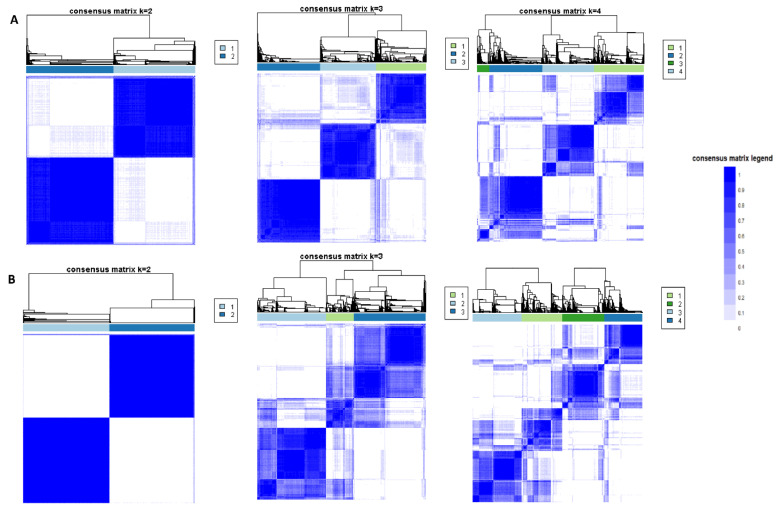
(**A**) Consensus matrix heat map displaying consensus values on a white to blue color scale for each cluster of patients with hypophosphatemia; (**B**) Consensus matrix heat map displaying consensus values on a white to blue color scale for each cluster of patients with hyperphosphatemia.

**Figure 2 jcm-10-04441-f002:**
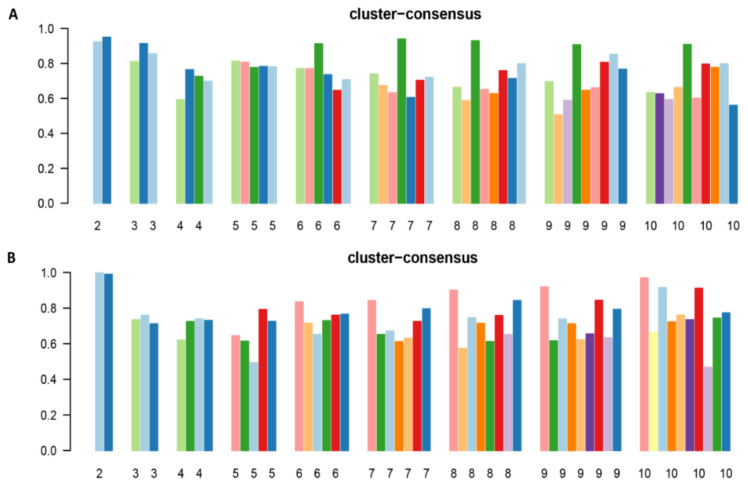
(**A**) Bar plots representing the mean consensus score for hypophosphatemic patients per different numbers of clusters (ranging from 2 to 10); (**B**) Bar plots representing the mean consensus score for hyperphosphatemic patients per different numbers of clusters (ranging from 2 to 10).

**Figure 3 jcm-10-04441-f003:**
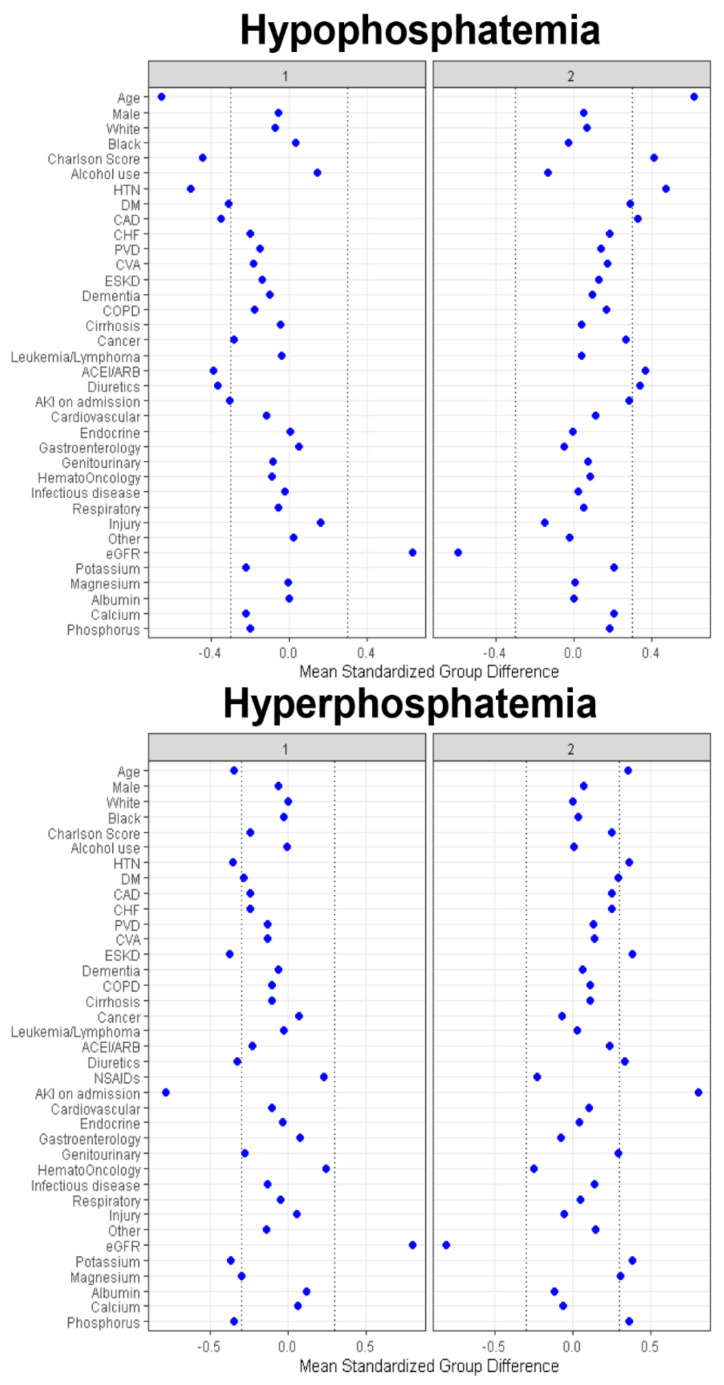
Standardized mean differences across the two clusters for each baseline variable for patients with hypophosphatemia and hyperphosphatemia. The x axis represents the standardized differences value, and the y axis represents baseline variables. The dashed vertical lines signify the standardized differences values of <−0.3 or >0.3. Abbreviations: AG, anion gap; AKI, acute kidney injury; BMI, body mass index; CHF, congestive heart failure; Cl, chloride; COPD, chronic obstructive pulmonary disease; CVA, cerebrovascular accident; DM, diabetes mellitus; ESKD, end stage kidney disease; GFR, glomerular filtration rate; GI, gastrointestinal; Hb, hemoglobin; HCO3, bicarbonate; K, potassium; ID, infectious disease; MI, myocardial infarction; Na, sodium; PVD, peripheral vascular disease; RS, respiratory system; SID, strong ion difference.

**Figure 4 jcm-10-04441-f004:**
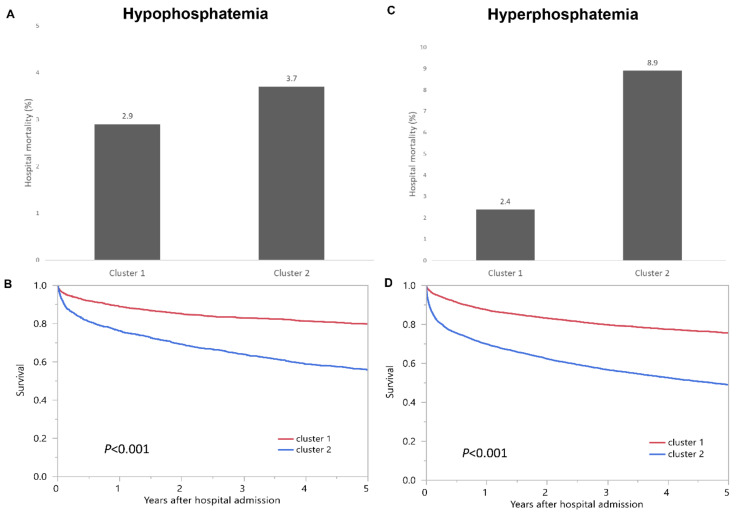
(**A**) Hospital mortality and (**B**) one-year and five-year mortality among two clusters of admission hypophosphatemia; (**C**) Hospital mortality and (**D**) one-year and five-year mortality among two clusters of admission hyperphosphatemia.

**Table 1 jcm-10-04441-t001:** Clinical characteristics.

Patient Characteristics	Hypophosphatemia Cohort				Hyperphosphatemia Cohort			
	Overall (*n* = 3113)	Cluster 1 (*n* = 1505)	Cluster 2 (*n* = 1608)	*p*-Value	Overall (*n* = 7252)	Cluster 1 (*n* = 3662)	Cluster 2 (*n* = 3590)	*p*-Value
Age (years)	60.6 ± 17.3	49.3 ± 14.8	71.2 ± 11.9	<0.001	59.7 ± 17.5	53.6 ± 17.1	66.0 ± 15.8	<0.001
Male sex	1652 (53)	759 (50)	893 (56)	0.004	3975 (55)	1890 (52)	2085 (58)	<0.001
Race				0.001				0.02
White	2849 (92)	1348 (90)	1501 (93)		6539 (90)	3303 (90)	3236 (90)	
Black	53 (2)	32 (2)	21 (1)		169 (2)	69 (2)	100 (3)	
Others	211 (7)	125 (8)	86 (5)		544 (8)	290 (8)	254 (7)	
Principal diagnosis				<0.001				<0.001
Cardiovascular	367 (12)	121 (8)	246 (15)		1214 (17)	472 (13)	742 (21)	
Endocrine/metabolic	151 (5)	75 (5)	76 (5)		407 (6)	173 (5)	234 (7)	
Gastrointestinal	562 (18)	302 (20)	260 (16)		892 (12)	542 (15)	350 (10)	
Genitourinary	78 (3)	19 (1)	59 (4)		799 (11)	80 (2)	719 (20)	
Hematology/oncology	459 (15)	175 (12)	284 (18)		1496 (21)	1118 (31)	378 (11)	
Infectious disease	359 (12)	163 (11)	196 (12)		340 (5)	68 (2)	272 (8)	
Respiratory	216 (7)	84 (6)	132 (8)		313 (4)	123 (3)	190 (5)	
Injury/poisoning	513 (16)	338 (22)	175 (11)		1051 (14)	600 (16)	451 (13)	
Other	408 (13)	228 (15)	180 (11)		740 (10)	486 (13)	254 (7)	
Charlson Comorbidity Score	2.1 ± 2.6	0.9 ± 1.5	3.1 ± 2.8	<0.001	2.5 ± 2.6	1.8 ± 2.4	3.1 ± 2.8	<0.001
Comorbidities								
Hypertension	1555 (50)	374 (25)	1181 (73)	<0.001	4189 (58)	1472 (40)	2717 (76)	<0.001
Diabetes mellitus	615 (20)	112 (7)	503 (31)	<0.001	2028 (28)	559 (15)	1469 (41)	<0.001
Coronary artery disease	507 (16)	50 (3)	457 (28)	<0.001	1558 (21)	422 (12)	1136 (32)	<0.001
Congestive heart failure	153 (5)	10 (0.7)	143 (9)	<0.001	722 (10)	99 (3)	623 (17)	<0.001
Peripheral vascular disease	87 (3)	5 (0.3)	82 (5)	<0.001	351 (5)	75 (2)	276 (8)	<0.001
Stroke	201 (6)	29 (2)	172 (11)	<0.001	532 (7)	139 (4)	393 (11)	<0.001
End-stage kidney disease	91 (3)	9 (0.6)	82 (5)	<0.001	1178 (16)	91 (2)	1087 (30)	<0.001
Dementia	46 (1)	4 (0.3)	42 (3)	<0.001	77 (1)	16 (0.4)	61 (2)	<0.001
COPD	298 (10)	65 (4)	233 (14)	<0.001	766 (11)	268 (7)	498 (14)	<0.001
Cirrhosis	147 (5)	58 (4)	89 (6)	0.03	284 (4)	69 (2)	215 (6)	<0.001
Cancer	782 (25)	193 (13)	589 (37)	<0.001	2014 (28)	1131 (31)	883 (25)	<0.001
Leukemia/lymphoma	242 (8)	101 (7)	141 (9)	0.03	385 (5)	172 (5)	213 (6)	0.02
Alcohol use	322 (10)	222 (15)	100 (6)	<0.001	415 (6)	206 (6)	209 (6)	0.72
Laboratory test								
eGFR (mL/min/1.73 m^2^)	82 ± 29	100 ± 22	64 ± 24	<0.001	55 ± 39	86 ± 27	23 ± 17	<0.001
Potassium (mEq/L)	3.9 ± 0.6	3.8 ± 0.6	4.0 ± 0.7	<0.001	4.5 ± 0.8	4.2 ± 0.6	4.8 ± 0.9	<0.001
Magnesium (mg/dL)	1.8 ± 0.3	1.8 ± 0.3	1.8 ± 0.4	0.73	2.0 ± 0.5	1.8 ± 0.3	2.1 ± 0.5	<0.001
Albumin (g/dL)	3.3 ± 0.5	3.3 ± 0.6	3.3 ± 0.4	0.89	3.4 ± 0.5	3.5 ± 0.5	3.3 ± 0.5	<0.001
Total calcium (mg/dL)	8.7 ± 0.9	8.5 ± 0.8	8.9 ± 1.0	<0.001	8.8 ± 0.8	8.9 ± 0.7	8.8 ± 0.9	<0.001
Phosphorus (mg/dL)	2.0 ± 0.4	2.0 ± 0.4	2.1 ± 0.3	<0.001	5.5 ± 1.2	5.0 ± 0.6	5.9 ± 1.5	<0.001
Medication								
ACEI/ARB	974 (31)	199 (13)	775 (48)	<0.001	2916 (40)	1062 (29)	1854 (52)	<0.001
Diuretics	1013 (33)	234 (16)	779 (48)	<0.001	3036 (42)	947 (26)	2089 (58)	<0.001
Acute kidney injury	501 (16)	74 (5)	427 (27)	<0.001	3358 (46)	259 (7)	3099 (86)	<0.001

**Table 2 jcm-10-04441-t002:** Mortality per cluster in hypophosphatemia and hyperphosphatemia.

	Hospital Mortality	OR (95% CI)	One-Year Mortality	HR (95% CI)	Five-Year Mortality	HR (95% CI)
(a) Hypophosphatemia cohort						
Cluster 1	2.9%	1 (ref)	14.0%	1 (ref)	20.2%	1 (ref)
Cluster 2	3.7%	1.32 (0.89–1.96)	26.8%	2.10 (1.75–2.52)	44.3%	2.56 (2.24–2.93)
(b) Hyperphosphatemia cohort						
Cluster 1	2.4%	1 (ref)	14.8%	1 (ref)	24.5%	1 (ref)
Cluster 2	8.9%	4.06 (3.18–5.17)	32.9%	2.63 (2.36–2.93)	51.1%	2.58 (2.38–2.79)

## Data Availability

Data is available upon reasonable request to the corresponding author.

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
