# Peer review of "Machine Learning Consensus Clustering Approach for Hospitalized Patients with Phosphate Derangements"

_jcm, 2021, doi:10.3390/jcm10194441_

Round 1
Reviewer 1 Report
Dear authors,
congratulations for your well conceived article, which is both scientifically sound and interesting.
There are just minor concerns on your manuscript
1) Methods: date were recorded from electronic hospital database, froma 2009 to 2013. Where follow-up date were collected? As machine analyzed both 1-year mortality and hospital mortality
2) What was the mean follow-up time available for patients? Did you stop collecting data after an year? In case you have further follow-up, what are 5-year mortality data?
3) In case further follow-up is lacking, it should be also stated in study limitations
Author Response
Response to Reviewer#1
Comment
congratulations for your well conceived article, which is both scientifically sound and interesting.
There are just minor concerns on your manuscript
Response: We thank you for reviewing our manuscript and for your critical evaluation.
Comment #1
Methods: date were recorded from electronic hospital database, froma 2009 to 2013. Where follow-up date were collected? As machine analyzed both 1-year mortality and hospital mortality
Response: The following statements have been added to the method section to describe the source of death data.
“The outcomes were hospital mortality, 1-year, and 5-year mortality. Patient death was obtained from our hospital’s registry and Social Security Death Index. The last follow-up date was December 31, 2018.”
Comment #2
What was the mean follow-up time available for patients? Did you stop collecting data after an year? In case you have further follow-up, what are 5-year mortality data?
Response: The following statements have been added to describe the median follow-up time in our study.
“The median follow-up date was 6.1 (IQR 1.8-8.0) years.”
In addition, we added 5-year mortality data in table 2. In briefly, cluster 2 had higher mortality than cluster 1 at 1 and 5 years in both hypophosphatemia and hyperphosphatemia cohort.
|
|
Hospital mortality |
OR (95% CI) |
1-year mortality |
HR (95% CI) |
5-year mortality |
HR (95% CI |
|
a) Hypophosphatemia cohort |
||||||
|
Cluster 1 |
2.9% |
1 (ref) |
14.0% |
1 (ref) |
20.2% |
1 (ref) |
|
Cluster 2 |
3.7% |
1.32 (0.89-1.96) |
26.8% |
2.10 (1.75-2.52) |
44.3% |
2.56 (2.24-2.93) |
|
b) Hyperphosphatemia cohort |
||||||
|
Cluster 1 |
2.4% |
1 (ref) |
14.8% |
1 (ref) |
24.5% |
1 (ref) |
|
Cluster 2 |
8.9% |
4.06 (3.18-5.17) |
32.9% |
2.63 (2.36-2.93) |
51.1% |
2.58 (2.38-2.79) |
Comment #3
In case further follow-up is lacking, it should be also stated in study limitations
Response: We appreciate reviewer’s important comment. We added the result of 5-year mortality, as mentioned in comment#2.
Thank you for your time and consideration. We greatly appreciated the reviewer’s and editor’s time and comments to improve our manuscript. The manuscript has been improved considerably by the suggested revisions.

Reviewer 2 Report
Abnormal phosphorus metabolism is a common condition in CKD patients and not only causes hyperparathyroidism and renal osteopathy, but its exacerbation leads to increased vascular calcification and increased cardiovascular events. Invite and increase the mortality rate. The data are generally well represented.
Author Response
Response to Reviewer
Comment
Abnormal phosphorus metabolism is a common condition in CKD patients and not only causes hyperparathyroidism and renal osteopathy, but its exacerbation leads to increased vascular calcification and increased cardiovascular events. Invite and increase the mortality rate. The data are generally well represented.
Response: We thank you for reviewing our manuscript and for your critical evaluation.
Thank you for your time and consideration. We greatly appreciated the reviewer’s and editor’s time and comments to improve our manuscript. The manuscript has been improved considerably by the suggested revisions.
